# The Uptake of Heparanase into Mast Cells Is Regulated by Its Enzymatic Activity to Degrade Heparan Sulfate

**DOI:** 10.3390/ijms25116281

**Published:** 2024-06-06

**Authors:** Jia Shi, Yoshiki Onuki, Fumiya Kawanami, Naoko Miyagawa, Fumika Iwasaki, Haruna Tsuda, Katsuhiko Takahashi, Teruaki Oku, Masato Suzuki, Kyohei Higashi, Hayamitsu Adachi, Yoshio Nishimura, Motowo Nakajima, Tatsuro Irimura, Nobuaki Higashi

**Affiliations:** 1Department of Biochemistry, Hoshi University School of Pharmacy, 2-4-41, Ebara, Shinagawa-ku 142-8501, Tokyo, Japan; d2181@hoshi.ac.jp (J.S.); s191058@hoshi.ac.jp (Y.O.); s171156@hoshi.ac.jp (H.T.); ka-takahashi@hoshi.ac.jp (K.T.); 2Department of Microbiology, Hoshi University School of Pharmacy, 2-4-41, Ebara, Shinagawa-ku 142-8501, Tokyo, Japan; oku@hoshi.ac.jp; 3Department of Clinical and Analytical Biochemistry, Faculty of Pharmaceutical Sciences, Tokyo University of Science, 2641, Yamazaki, Noda 278-8510, Chiba, Japanhigase@rs.tus.ac.jp (K.H.); 4Institute of Microbial Chemistry (BIKAKEN), 18-24, Miyamoto, Numazu 410-0301, Shizuoka, Japan; adachih@bikaken.or.jp; 5Institute of Microbial Chemistry (BIKAKEN), 3-14-23, Kamiosaki, Shinagawa-ku 141-0021, Tokyo, Japan; nyoshiocin@jcom.home.ne.jp; 6SBI Pharmaceuticals Co., Ltd., 1-6-1, Roppongi, Minato-ku 106-6019, Tokyo, Japan; motnakaj@sbigroup.co.jp; 7Division of Glycobiologics, Juntendo University Graduate School of Medicine, 2-1-1, Hongo, Bunkyo-ku 113-8421, Tokyo, Japan; t-irimura@juntendo.ac.jp

**Keywords:** endocytosis, heparanase, heparan sulfate degradation, heparastatin (SF4), heparin, mast cells, syndecan

## Abstract

Mast cells take up extracellular latent heparanase and store it in secretory granules. The present study examined whether the enzymatic activity of heparanase regulates its uptake efficiency. Recombinant mouse heparanase mimicking both the latent and mature forms (L-Hpse and M-Hpse, respectively) was internalized into mastocytoma MST cells, peritoneal cell-derived mast cells, and bone marrow-derived mast cells. The internalized amount of L-Hpse was significantly higher than that of M-Hpse. In MST cells, L-Hpse was continuously internalized for up to 8 h, while the uptake of M-Hpse was saturated after 2 h of incubation. L-Hpse and M-Hpse are similarly bound to the MST cell surface. The expression level of cell surface heparan sulfate was reduced in MST cells incubated with M-Hpse. The internalized amount of M-Hpse into mast cells was significantly increased in the presence of heparastatin (SF4), a small molecule heparanase inhibitor that does not affect the binding of heparanase to immobilized heparin. Enzymatically quiescent M-Hpse was prepared with a point mutation at Glu335. The internalized amount of mutated M-Hpse was significantly higher than that of wild-type M-Hpse but similar to that of wild-type and mutated L-Hpse. These results suggest that the enzymatic activity of heparanase negatively regulates the mast cell-mediated uptake of heparanase, possibly via the downregulation of cell surface heparan sulfate expression.

## 1. Introduction

Heparanase (Hpse) has been identified as a sole heparan sulfate (HS) degradation enzyme that generates fragmented oligosaccharide HS composed of 10–20 disaccharide units in mammals [1]. Hpse is initially synthesized as a latent enzyme that lacks enzymatic activity while possessing binding capacity to heparin, a highly sulfated glycosaminoglycan (GAG) with a backbone structure common to HS [2,3,4,5]. Removal of the intervening peptide by cathepsin L-mediated cleavage generates a mature enzyme as a heterodimer composed of 8-kDa and 46-kDa subunits in mice that cleaves HS and macromolecular heparin [6,7]. In addition to the enzymatic activity, Hpse can transduce cellular signals to induce angiogenesis, cell spreading, chemokine expression, etc. Enzymatic activity is dispensable for the functions, i.e., the latent-form Hpse can bind to cell surface receptors for further signal transduction [8,9,10,11,12,13,14]. In this sense, the molecular form of Hpse critically defines the spectrum of Hpse-mediated biological activities. 

Hpse upregulation has been reported in many inflammatory diseases, including delayed-type hypersensitivity, psoriasis, inflammatory bowel disease, rheumatoid arthritis, atherosclerosis, fibrosis, pulmonary inflammation, pancreatitis, acute kidney injury, etc. [15,16,17,18,19,20,21,22]. Comprehensive gene expression analysis shows relatively high expression of Hpse in leukocytes, including mast cells, which has been confirmed by us and other researchers [23,24,25,26]. In addition, epithelial cells can produce Hpse in inflammatory diseases.

Mast cells trigger allergic and inflammatory reactions by releasing various vasoactive and inflammatory substances. Secretory granules of mast cells contain a wide spectrum of substances, such as vasoactive amines, cytokines, lysosomal enzymes, etc. A large part of them are preformed and stored in the granules. In addition, mast cells can take up extracellular materials and transfer them into the secretory granules. The uptake process is required for the maturation and activation of the internalized proteins. As an example, internalized TNF-alpha becomes mature during the uptake process [27,28]. 

Hpse is essential for heparin processing in mast cells. It is likely that mast cells produce latent-form Hpse, secrete it, and then take it up and convert it into mature-form Hpse for use. Since mast cells release the mature-form Hpse as a result of degranulation, replenishment of the enzymatically active Hpse in the granules is essential for maintaining continuous heparin processing. Our previous study indicated that recombinant latent-form Hpse can be internalized and sorted into secretory granules. The enzyme was processed into mature-form Hpse during the internalization process [29,30]. Additionally, this process is relevant for replenishing granular contents that have been released via degranulation. In this case, endocytosis occurring coordinately with the degranulation has a role in the replenishment [31]. It has been reported that mast cells can internalize IL-17, MHC class II, PSGL-1, and IgE fragments presumably associated with FcεRI [32,33,34,35,36]. Despite these studies, the molecular mechanism of the endocytic process in mast cells has not been intensively studied. 

Accumulated information suggests the involvement of Hpse in mast cell biology for at least two functional aspects. One aspect is cellular trafficking. Degranulating mast cells secrete Hpse, which can cleave HS deposited in the basement membrane [24]. A potential role of Hpse in transendothelial migration has been suggested. Mice genetically deficient in Hpse showed a lower efficiency in cellular trafficking of eosinophils, monocytes, and dendritic cells [22,37,38,39,40]. Topical administration of a Hpse inhibitor suppressed the extravasation of leukocytes into the inflamed tissues [41], which is promising for future pharmaceutical application of Hpse inhibitors in inflammatory diseases. Although mast cells also migrate into inflammatory lesions in diseases such as allergic airway inflammation and atherosclerosis [42,43,44], the involvement of Hpse in mast cell trafficking has not been intensively studied. 

The other aspect is the processing of granular heparin, a highly sulfated GAG specifically accumulated in the secretory granules of mast cells. Heparin is initially synthesized as macromolecular heparin comprising heparin chains attached to a core protein, serglycin [45,46]. Hpse-mediated cleavage liberates the heparin chain from the core protein and generates fragmented heparin with a molecular weight of around 10 kDa [26,47]. We have proposed that this Hpse-mediated cleavage accelerates the release of granular components of mast cells from extracellular matrices [29]. Other researchers indicated that Hpse expression results in decreased storage amounts of granular enzymes such as carboxypeptidase A and mMCP-5 [26], as well as in the increased production of total GAGs [48]. 

The above two aspects are related to the expression level and maturation status of Hpse in secretory granules. Our immunostaining studies indicated abundant expression of Hpse in mast cells in the peritoneal cavity, and in the skin; however, mature mast cells obtained in in vitro culture by differentiation with cytokines did not always express Hpse in the granules [29]. These results raised the question as to how mast cells obtain functional Hpse in the secretory granules. Our precedent study proposed a possible pathway, that is, mast cells take up extracellular latent-form Hpse and store it in their secretory granules. During this process, the internalized Hpse was processed into mature-form Hpse that can degrade macromolecular heparin in the secretory granules into fragments. Syndecan 4 (Sdc4) on the mast cell surface is considered essential and the dominant surface molecule for Hpse uptake [30]. The aforementioned study, however, did not mention whether the uptake process merely replenishes the granular enzyme or whether it is associated with the maturation of Hpse. Consequently, in the present study we focus on a putative regulatory system of the uptake process. Using four types of recombinant Hpse proteins, we present evidence showing that Hpse enzymatic activity has a role in the attenuation of Hpse uptake. 

## 2. Results

### 2.1. Recombinant Latent-Form Hpse Was Efficiently Internalized into Mast Cells

Recombinant mouse Hpse protein mimicking the latent and the mature form (hereafter termed L-Hpse and M-Hpse, respectively) was prepared [14,23,30,49]. The recombinant Hpse protein was incubated with MST cells, peritoneal cell-derived mast cells (PMC), and bone marrow-derived mast cells (BMMC). The uptake was more efficient with L-Hpse (Figure 1A–C). The time course of the uptake of Hpse into MST cells was further studied. Uptake of L-Hpse was continuous at least up to 8 h, whereas uptake of M-Hpse was less efficient and saturated after 2 h of incubation (Figure 1A,D). Immunocytochemical images showed that the fluorescence intensity was higher in MST cells incubated with L-Hpse than in MST cells incubated with M-Hpse. The L-Hpse-dependent granular fluorescent signal was strong, particularly in the intracellular area surrounding the nucleus of MST cells (Figure 1E).

### 2.2. Incubation with M-Hpse Decreased HS Expression on the Cell Surface of MST

We examined whether the expression level of the HS chain was influenced as a result of incubation with M-Hpse. MST cells treated with M-Hpse were weakly stained with an anti-HS antibody 10E4, suggesting that the expression level of cell surface HS decreased after the incubation with M-Hpse. The expression level of Sdc4 was equivalent in untreated MST cells and those treated with L-Hpse and M-Hpse; therefore, the decrease in cell surface HS expression was likely not due to the downregulation of a core protein (Figure 2). The binding of Hpse to the cell surface of MST was detectable as an increase in the median fluorescence intensity (MFI) using the anti-muHpse mAb-based fluorescent signal. The cell surface binding of L-Hpse and M-Hpse gave rise to an increase in MFI as 1.13 ± 0.06-fold (L-Hpse, *n* = 3) and 1.14 ± 0.10-fold (M-Hpse, *n* = 3), respectively, suggesting that the initial binding of the Hpse was comparable for L-Hpse and M-Hpse.

### 2.3. The Uptake of M-Hpse Was Enhanced in the Presence of Heparastatin (SF4)

The uptake was further examined in the presence of heparastatin (SF4), a small molecule inhibitor of Hpse [50]. The uptake of M-Hpse into MST cells was enhanced in the presence of heparastatin (SF4) to a similar extent as the uptake of L-Hpse (Figure 3A). Similar enhanced uptake was also shown in PMC and BMMC, where the uptake of M-Hpse as well as L-Hpse was evident (Figure 3B,C). Heparastatin (SF4) at a concentration of 1 mM did not affect the binding of M-Hpse to immobilized heparin (Figure 3D).

### 2.4. M-Hpse with a Point Mutation Efficiently Internalized into MST Cells

To further elucidate the regulatory role of the enzymatic activity of Hpse in the uptake, four types of Hpse recombinant proteins were prepared, namely L-Hpse and M-Hpse with wild-type amino acid sequences (L-Hpse wt and M-Hpse wt, respectively), and their mutated forms (L-Hpse mut and M-Hpse mut, respectively) with substitution of Glu335 with alanine (Figure 4A). The latter two lacked enzymatic activity (Figure 4B and Appendix A). The uptake efficiency of M-Hpse mut was similar to that of L-Hpse wt and L-Hpse mut and significantly higher than that of M-Hpse wt (Figure 4C). Binding to immobilized heparin and to GAG derived from MST cells was mostly equivalent in the four types of recombinant proteins (Figure 4D,E and Appendix A, Appendix A).

## 3. Discussion

An essential question in the present study was whether the uptake process of Hpse merely replenishes the granular enzyme or whether it is always associated with the maturation of Hpse. If the latter is the case, the underlying mechanism of how L-Hpse is preferentially internalized becomes an additional question. The current study was designed to examine whether different molecular forms of Hpse are similarly internalized into mast cells. The uptake of L-Hpse was more efficient than that of M-Hpse in MST mastocytoma cells, PMC, and BMMC, where the uptake of M-Hpse was saturated earlier, at 2 h of incubation. The next question was whether M-Hpse cleaved GAG in mast cells because M-Hpse is enzymatically active. Flow cytometric analysis indicated relatively weak staining of M-Hpse-treated MST with an anti-HS antibody 10E4, suggesting that the amount of cell surface HS was decreased without affecting the expression level of Sdc4. The involvement of Hpse enzymatic activity was confirmed by using a heparanase inhibitor, heparastatin (SF4), and enzymatically quiescent mutated Hpse proteins. Taken together, we propose a regulatory role of Hpse enzymatic activity for the preferential uptake of L-Hpse. 

In a previous study, we examined the time course of the uptake of L-Hpse in MST cells. The uptake was continuously increased up to 6 h and then saturated at 24 h. Processing of the internalized L-Hpse into the mature form was detectable at 6 h and mostly completed at 14 h [30]. The saturation may be due to a limited storage capacity in the granules or a regulatory system to avoid excessive accumulation of Hpse. Our current study suggests that the mature enzyme can negatively regulate the uptake process because inhibition of enzymatic activity using heparastatin (SF4) or addition of enzymatically quiescent M-Hpse mut fully restore the uptake efficiency. It is still not clear how PMC and BMMC showed enhanced uptake efficiency of L-Hpse as well as M-Hpse in the presence of heparastatin (SF4). It is speculated that mature Hpse generated as a result of intracellular processing of the internalized L-Hpse could have a regulatory role, which will be the subject of clarification in a future study. 

MST cells incubated with M-Hpse reduced the expression of HS on the cell surface, suggesting Hpse degraded the HS that was detected on the cell surface by flow cytometry. Because Hpse is enzymatically inactive under neutral pH (~7) [4], it is expected that the degradation occurs not on the cell surface but under acidic conditions (pH 5~6), such as in early or late endosomes where M-Hpse can be transferred. In addition to the decrease in the amount of HS, Hpse-mediated degradation may alter the disaccharide composition of HS, which has been reported in a previous study showing that disaccharide composition in HS was altered in Hpse transgenic mice [51]. Sdc4 gene knockout experiments in our previous study strongly suggested the involvement of Sdc4 as a main receptor for the uptake of L-Hpse. However, since approximately 50% of the uptake remained after knockdown, we concluded that other cell surface GAGs, such as Sdc1 and Sdc3, weakly expressed in MST cells [30] are also possibly involved in the uptake. 

Heparastatin (SF4) enhanced the uptake of M-Hpse without affecting the binding of M-Hpse to immobilized heparin (Figure 3). Because the uptake of L-Hpse in MST was not influenced by the presence of heparastatin (SF4), it is not likely that heparastatin (SF4) stimulates the cells to accelerate the uptake process. Hpse carries two heparin-binding domains (HBDs) on the surface of the molecule, HBD-1 and -2, which are positively charged with lysine and arginine residues inside [52]. It is likely that electrostatic interaction is dominant in the initial enzyme-substrate interaction. Glu335 acting as a nucleophile, coupled with another glutamate residue (Glu217) as a proton donor, is an essential amino acid residue for the enzymatic activity of mouse Hpse. A crystallographic study of Hpse demonstrated the location of these anionic amino acid residues relatively inside of the enzyme molecule [53]. It is speculated that the negative charge itself can be repulsive for the binding of Hpse to HS, which is also supported by our previous physicochemical study [54]. Heparastatin (SF4), a compound optimized for inhibition of Hpse enzymatic activity, is a derivative of a bacterial product siastatin B that can inhibit the enzymatic activity of Hpse. The complex structure of human Hpse and siastatin B was recently demonstrated using 3D crystallography [55]. That study identified amino acid residues that are involved in the interaction of Hpse with siastatin B. These are Asp62, Thr97, and glutamic acids required for the enzymatic activity (Glu225 and Glu343), none of which are cationic residues located in HBD-1 or -2. Therefore, it is not likely that the interaction of Hpse with siastatin B influences the initial electrostatic interaction with HS. In other words, the mode of binding of siastatin B suggests that the binding sites and the cleavage sites of HS are segregated in the Hpse molecule. 

To further examine the regulatory role of Hpse enzymatic activity for the uptake, mutant proteins with substitution of Glu335 with alanine were prepared. Glu335 initiates the hydrolysis reaction of HS by attacking the glycosidic bond of glucuronic acids [56]. A point mutation at Glu343 in human Hpse, corresponding to Glu335 in mice, resulted in a complete loss of enzymatic activity [56]. We confirmed a similar loss of Hpse activity in the M-Hpse mut (Figure 4B). Substitution of Glu335 into alanine in L-Hpse and M-Hpse did not affect the binding of Hpse to immobilized heparin, suggesting that the binding to cell surface receptors was scarcely influenced by the point mutation. Relatively efficient internalization of M-Hpse mut again strongly suggests that the enzymatic activity is a key regulatory mechanism for the uptake efficiency.

The preferential uptake of latent-form Hpse likely facilitates efficient maturation of Hpse. Although the further physiological relevance of the uptake of Hpse remains to be elucidated, implications of the preferential uptake could be discussed, focusing on three points. First, the uptake system is beneficial to avoid exaggerated inflammation induced by latent-form Hpse [10,14,57]. In other words, scavenging of latent-form Hpse may be an anti-inflammatory action by mast cells. Second, because there are many kinds of biologically active molecules that can bind to heparan sulfate, the mature-form Hpse-mediated regulation is also possibly suppressive for the uptake of other HS-associated molecules. We are currently working on clarifying this point by focusing on the heparin-binding cytokines FGF and VEGF that are stored in the secretory granules of mast cells [58,59,60,61]. Third, the presence of mature-form Hpse in the extracellular space may confer unresponsiveness to mast cells. We and others reported that secretory granules accumulate mature-form Hpse [23,24,25,26]. When this Hpse is released, degradation of cell surface HS by the released Hpse possibly retards the uptake process of mast cells. If such degradation did not occur, mast cells could rapidly re-internalize the released materials, which may decline mast cell-derived inflammatory responses. It is interesting that this process can be pharmacologically accelerated by the addition of Hpse inhibitors like heparastatin (SF4) (Figure 3). Further studies on the manipulation of mast cell-mediated uptake are ongoing. 

## 4. Materials and Methods

### 4.1. Reagents and Cells

Heparin from porcine intestinal mucosa (H7005, approx. 15 kDa), Triton X-100, bovine serum albumin (BSA) solution (1%) were purchased from Sigma (St. Louis, MO, USA); 1:1 mixture of D-MEM/Ham’s F-12 medium (D/F) and Tween-20 from Wako Pure Chemical (Tokyo, Japan); CHAPS from Nacalai tesque (Kyoto, Japan); heparan sulfate from Seikagaku (Tokyo, Japan); normal goat serum from Japan Laboratory Animals, Inc. (Tokyo, Japan). The heparanase inhibitor heparastatin (SF4) was synthesized by Dr. Hayamitsu Adachi, as shown elsewhere [50]. The following antibodies were used: anti-HS (clone F58-10E4) from Amsbio (Madrid, Spain), anti-syndecan-4 (clone KY/8.2) from Pharmingen (San Diego, CA, USA), rat IgG2a from eBiosciences (San Diego, CA, USA), mouse IgM and HRP-conjugated goat anti-rabbit IgG(H+L) from Zymed (San Francisco, CA, USA). Antibody binding was visualized after a 3-step staining procedure using biotinylated goat anti-rat IgG(H+L), FITC- or horseradish peroxidase-conjugated streptavidin (Zymed). Anti-muHpse mAb RIO-1 and rabbit anti-muHpse antiserum were prepared as stated elsewhere [23]. A connective tissue-type mastocytoma cell line MST was kindly provided by Prof. Esko [62] and maintained in a D/F medium containing 10% FCS. PMC and BMMC were generated, as shown elsewhere [29]. The animal protocols were approved by the Animal Care and Use Committee of Hoshi University School of Pharmacy and Pharmaceutical Sciences (No. 29-064).

### 4.2. Preparation of Recombinant Hpse Protein

Baculovirus for the expression of Hpse was prepared following the manufacturer’s protocol. The supernatant of infected Sf9 insect cells was collected and purified as described elsewhere [14,49]. To prepare highly purified mutated proteins, the C terminus 6 × His tag attached with a spacer sequence GGGGS was inserted by PCR of the original vectors (pFastbac_muHpse and pFastbac Dual_muHpse) using KOD (Toyobo, Osaka, Japan) and the primers given below.

pFastbac forward: CAAAATTGCTGCTTGTATAggtggaggtggatcccatcatcatcaccaccacTGAAAGCTTGTCGAGAAGTACTAGAG 

pFastbacDual forward: CAAAATTGCTGCTTGTATAggtggaggtggatcccatcatcatcaccaccacTGACCATGGTGCTAGCAGC

pFastbac/pFastbacDual reverse: 

tggtgatgatgatgggatccacctccaccTATACAAGCAGCAATTTTGGCATTTCTTATG. 

The alanine mutation at Glu335 (E335A) was further inserted into the above vectors using similar KOD-based PCR with the primers given below.

E335A forward: AGAAGGTCTGGTTGGGAGcGACGAGCTCAGCTTACGG

E335A reverse: CCRTAAGCTGAGCTCGTCgCTCCCAACCAGACCTTCT

Insertion of these mutations was confirmed by nucleotide sequencing (Fasmac Co., Ltd., Kanagawa, Japan). The supernatant of Sf9 cells that had been infected with reconstituted baculovirus (20 mL) was collected, diluted with an equal volume of buffer A (50 mM Tris-HCl buffer (pH 7.6), and subjected to an NTA-Ni Sepharose column (ϕ 8 × 10 mm, Fujifilm-Wako, Osaka, Japan). After washing with buffer A containing 20 mM imidazole, recombinant protein bound to the resin was eluted with buffer A containing 250 mM imidazole. 

### 4.3. Quantification of Hpse Uptake in Mast Cells

The uptake of Hpse inside the mast cells was quantitatively determined, as reported elsewhere [29,30]. In brief, MST cells (2 × 10^5^ cells), PMC, or BMMC (1 × 10^5^ cells) were seeded in D/F medium containing 1% BSA (Sigma-Aldrich, Burlington, MA, USA, B1111) in a well of a 24-well plate or 96-well plate (Sumitomo Bakelite (Tokyo, Japan) specified for non-adhesive cells, MS-8024R and MS-8096R). A known concentration of Hpse recombinant protein was added into the medium, and the microtiter plates were kept at 37 °C. The Hpse inhibitor heparastatin (SF4) was preincubated with the recombinant proteins 30 min before addition to the plate. At different time points after the incubation, the cellular pellets were recovered in an Eppendorf tube, washed three times with ice-cold PBS, and lysed in 1% Triton X-100 in PBS for 30 min on ice. The lysates were subjected to sandwich ELISA as described elsewhere [29,30] and quantified as the concentration of Hpse in the cell lysates (ng/mL). 

### 4.4. Immunocytochemistry

MST cells were incubated with 400 ng/ml of Hpse protein for 8 h at 37 °C. After the incubation, the cell suspension was subjected to cytospin, fixed with ice-cold methanol for 30 s, stained with anti-muHpse mAb RIO-1, secondary antibody (biotinylated goat anti-rat IgG(H+L), Invitrogen, Carlsbad, CA, USA), and Alexa568-conjugated streptavidin. The fluorescent cell image was taken using confocal microscopy (FV3000, Olympus, Tokyo, Japan). 

### 4.5. Flow Cytometry

MST cells (4 × 10^5^ cells) were suspended in a D/F medium containing 1% BSA (400 µL) and incubated with 400 µg/ml of Hpse protein for 120 min at 37 °C. After the incubation, the cell suspension was quickly cooled on ice, and sodium azide solution (0.1% final) was added and washed to remove Hpse from the supernatant. The cells were divided into aliquots, immunostained with anti-HS or anti-Sdc4 antibodies, and analyzed with an Epics XL (Beckman Coulter, Brea, CA, USA) or FACS Verse (Becton Dickinson, Franklin Lakes, NJ, USA). 

To detect the binding of Hpse to the cell surface, MST cells (1 × 10^5^ cells) were suspended in PBS containing 3% BSA and 2% normal goat serum with sodium azide for blocking. Thereafter, the cells were mixed with Hpse (5 µg/mL) for 60 min at 4 °C and washed two times. Hpse bound on the cell surface was detectable by immunostaining using anti-muHpse mAb (RIO-1) and FITC-labeled goat anti-rat IgG(H+L). 

### 4.6. Binding of Hpse to Immobilized GAG

The binding of Hpse to immobilized heparin or GAG derived from MST cells was examined based on an ELISA-like method as described previously [49,54]. Briefly, streptavidin was immobilized onto the ELISA plate (Greiner, Frickenhausen, Germany, 655061). After blocking with 1% BSA-PBS, biotinylated GAG (10 µg/mL) in dilution buffer (PBS containing 0.1% BSA and 0.01% Tween 20) was added into each well. After washing out unbound GAG, Hpse was added into the wells and kept for 2 h at room temperature. In some experiments, Hpse was preincubated with heparastatin (SF4) before addition to the ELISA plate. Dilution buffer with or without heparastatin (SF4) was used as a negative control. The binding of Hpse was quantified by the coloration of ABTS at 405 nm. OD405 readings were normalized by subtracting the negative control OD values. The OD405 value of the negative control wells was 0.050 ± 0.003 (average ± S.D.). 

### 4.7. Measurement of HS Degradation Activity of Hpse

The HS degradation activity of each Hpse recombinant protein was detected using Superdex™ 75 Increase (5/150 GL, Cytiva, Uppsala, Sweden) as described elsewhere [14,23]. 

### 4.8. Preparation of MST-Derived GAG

The cell pellets of MST were lyophilized and digested with actinase E (Kaken Pharmaceutical Co., Ltd., Tokyo, Japan) at 37 °C for 48 h. For beta elimination, the digested materials were treated with 0.5 M NaOH and 0.15 M NaBH_4_ at 4 °C for 18 h. After neutralization with acetic acid to adjust to pH 7~8, the digested materials were diluted with the same volume of 5 M Urea solution and applied to a DEAE-Celluofine column (Seikagaku). The bound materials were washed with 50 mM Tris-HCl, 150 mM NaCl, 0.5% Triton X-100, 4 M Urea (pH 7.6), and then with 50 mM sodium acetate buffer, 150 mM NaCl, 0.5% Triton X-100, and 4 M Urea (pH 4.0). Afterward, GAG was eluted with 100 mM sodium acetate buffer, 1.5 M NaCl, 0.5% CHAPS (pH 5.3), and precipitated with 70% ethanol. The GAG was recovered as a precipitate of centrifugation (15,000 rpm, 30 min), solubilized with water, dialyzed, and lyophilized. The carboxyl group in the GAG structure was biotinylated using biotin-PEG4-hydrazide (B5578, Tokyo Chemical Industry, Tokyo, Japan) and N-(3-Dimethylaminopropyl)-N′-ethylcarbodiimide hydrochloride (E6383, Sigma-Aldrich) as stated elsewhere [49,54]. Disaccharide analysis [57,63] of the prepared GAG derived from MST cells indicated the composition as follows: ΔDi-0S, 31.1%; ΔDi-NS, 16.0%; ΔDi-6S, 7.4%; ΔDi-NS6S, 11.1%; ΔDi-2SNS, 2.8%; ΔDi-TriS, 31.6%. 

### 4.9. Statistical Analysis

The significance of differences in the data was evaluated using the two-tailed Student’s *t*-test. 

## 5. Conclusions

In the present study, we propose a novel role for the enzymatic activity of Hpse to regulate the uptake efficiency of Hpse into mast cell granules. This regulatory system is potentially significant for accelerating the maturation of the Hpse enzyme and for scavenging latent-form Hpse from inflammatory environments. Our findings imply that mature-form Hpse-mediated suppression of the uptake can be manipulated by using heparanase inhibitors, which can potentially provide a novel strategy to regulate extracellular inflammatory conditions. 

## Figures and Tables

**Figure 1 ijms-25-06281-f001:**
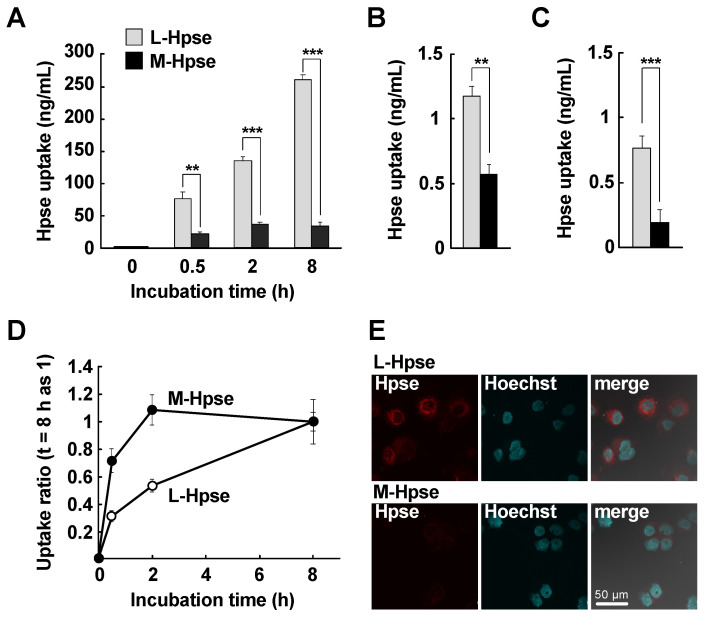
Uptake of L-Hpse and M-Hpse into mast cells. (**A**) MST cells were incubated with L-Hpse (gray bars) and M-Hpse (black bars) (400 ng/mL) for the indicated periods. (**B**,**C**) PMC (**B**) and BMMC (**C**) were incubated with L-Hpse (gray bars) and M-Hpse (black bars) (500 ng/mL) for 6 h. In (**A**–**C**), Hpse concentration in cell lysates was determined by ELISA. (**D**) The amount of Hpse at 8 h was defined as 100%, and the uptake ratio of L-Hpse (open circles) and M-Hpse (closed circles) is shown at each time point. Data are shown as mean ± S.D. **, ***: Significantly higher than control (**: *p* < 0.01; ***: *p* < 0.001). *n* = 3. (**E**) MST cells were incubated with L-Hpse and M-Hpse (400 ng/mL) for 8 h. Cell images immunostained with anti-Hpse mAb (RIO-1), secondary antibody, and Alexa 568-labeled streptavidin (Hpse). The cells were counterstained with Hoechst33342 (Hoechst). The right panels are merged figures of Hpse, Hoechst, and transmission images.

**Figure 2 ijms-25-06281-f002:**
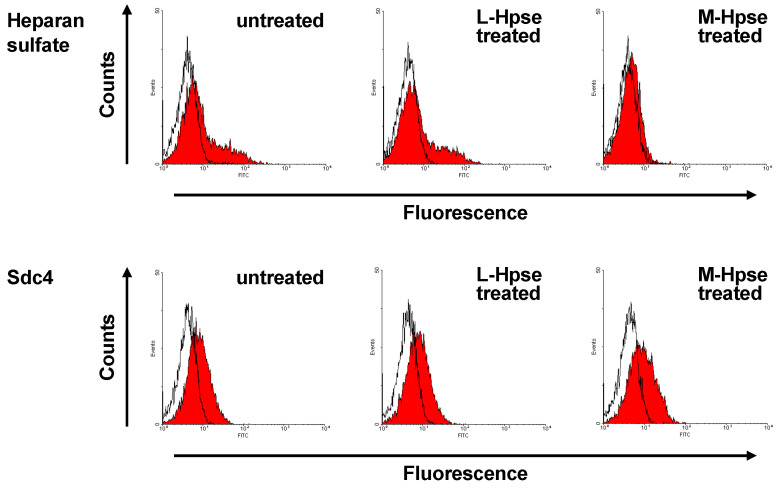
Expression level of HS and Sdc4 on Hpse-treated MST cells. Expression level of HS and Sdc4 was examined in MST cells either untreated or treated with Hpse recombinant proteins (400 ng/mL), which is shown in red. Staining with isotype control is shown in white.

**Figure 3 ijms-25-06281-f003:**
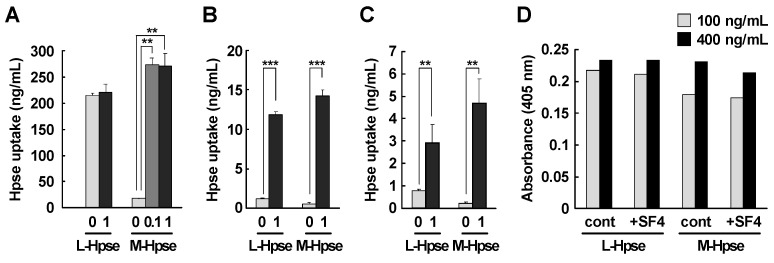
Enhanced uptake of M-Hpse Hpse into MST cells in the presence of heparastatin (SF4). (**A**–**C**) MST cells (**A**), PMC (**B**), and BMMC (**C**) were incubated with L-Hpse and M-Hpse (MST: 400 ng/mL, PMC, and BMMC: 500 ng/mL) pretreated with heparastatin (SF4) at the indicated concentrations (0.1 mM or 1 mM) for 6 h. Data are shown as mean ± S.D. **: Significantly higher than control (**: *p* < 0.01; ***: *p* < 0.001). *n* = 3. (**D**) Binding of L-Hpse and M-Hpse to immobilized heparin. Biotinylated heparin (10 µg/mL) was immobilized on an ELISA plate that was pretreated with 10 µg/mL streptavidin. Hpse at a concentration of 100 ng/mL (gray bars) or 400 ng/mL (closed bars) was either pretreated with 1 mM of heparastatin (SF4) for 30 min (+SF4) or untreated (cont). These samples were added to the ELISA plate. The binding of Hpse was quantified using the coloration of ABTS at 405 nm.

**Figure 4 ijms-25-06281-f004:**
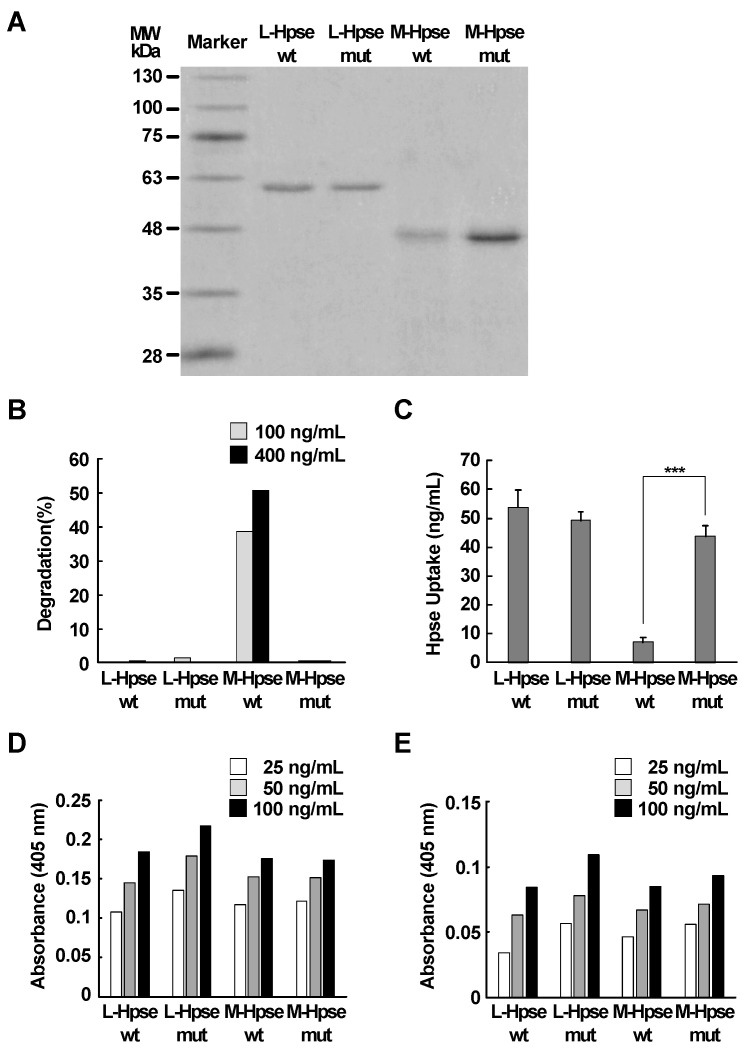
Uptake of wild-type and mutant Hpse into MST cells. (**A**) SDS-PAGE of purified recombinant Hpse (L-Hpse wt, L-Hpse mut, M-Hpse wt and M-Hpse mut). (**B**) Enzymatic activity of the purified recombinant proteins at 100 ng/mL (gray bars) or 400 ng/mL (closed bars) was measured. (**C**) Uptake of the four recombinant Hpse proteins into MST cells. MST cells were incubated with the protein (200 ng/mL) for 6 h. Data are shown as mean ± S.D. ***: Significantly lower than control (***: *p* < 0.001). *n* = 4. (**D**,**E**) Binding of the four recombinant Hpse proteins to immobilized heparin (**D**) or GAG derived from MST cells (**E**), 25 ng/mL (open bars), 50 ng/mL (gray bars), and 100 ng/mL (closed bars).

## Data Availability

The original contributions presented in the study are included in the article and Appendix A. Further inquiries can be directed to the corresponding author.

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
