# Peer review of "The Uptake of Heparanase into Mast Cells Is Regulated by Its Enzymatic Activity to Degrade Heparan Sulfate"

_ijms, 2024, doi:10.3390/ijms25116281_

Round 1

Reviewer 1 Report

Comments and Suggestions for Authors

The manuscript entitled "Uptake of heparanase into mast cells is regulated by its enzymatic activity to degrade heparan sulfate" by Shi et al. explores the impact of heparanase enzymatic activity on its uptake efficiency into mast cells, building upon previous research investigating intracellular trafficking and processing of extracellular heparanase by mast cells (see reference 28 in the manuscript). The topic addressed is both interesting and significant, and the experiments are well-designed with adequate methods. The paper is informative, well-written, and supported by data that effectively contribute to the conclusions drawn. Additionally, the work is well-referenced.

However, there are a few points that could enhance the manuscript:

1.     In Figure 1, the investigation of L-Hpse and M-Hpse uptake into mast cells using ELISA is pivotal. To bolster the reliability of the findings, it would be beneficial to confirm the results with alternative methods such as immunofluorescent microscopy utilizing an anti-heparanase antibody. Additionally, visualizing the localization of heparanase in endosomes through colocalization experiments using immunofluorescent microscopy could provide valuable insights.

2.     There appear to be errors in the figure legend for Figure 1 (Page 3, lines 95-97): (A-C) MST cells were incubated with L- 95 Hpse (gray bars) and M-Hpse (black bars) (400 ng/ml) for the indicated periods, (B, C) PMC and BMMC were incubated with Hpse (500 ng/ml) for 6 h!!!!

3.     In Figure 3D, it is unclear how the control experiments were conducted, and this should be addressed for clarity and completeness.

4.     In the discussion section, it would be beneficial to discuss the potential role of other cell surface proteoglycans in the uptake of heparanase, as this could provide further insights into the mechanisms involved.

Overall, addressing these points would strengthen the manuscript and contribute to a more comprehensive understanding of the topic.

Author Response

May 29, 2024

Response to Reviewer 1 Comments

[IJMS] Manuscript ID:  ijms-3025655 : major revisions
manuscript ID:  ijms-3025655
Type of manuscript: article
Title: Uptake of heparanase into mast cells is regulated by its enzymatic activity to degrade heparan sulfate
Section: Macromolecules
Submitted to special issue: Heparin, Heparan Sulfate and Heparanase in Health and Disease

Dear Editors,

First of all, let me express my sincere thanks for taking the time to review our manuscript and for your encouragement to submit a revised version. We carefully read the Reviewers` comments and hereby submit our point-by-point responses, including an explanation on how the manuscript was revised. In the revised manuscript, major revisions are indicated in red font. We hope that the revisions are satisfactory for the Reviewers.

Reviewer 1:
The manuscript entitled "Uptake of heparanase into mast cells is regulated by its enzymatic activity to degrade heparan sulfate" by Shi et al. explores the impact of heparanase enzymatic activity on its uptake efficiency into mast cells, building upon previous research investigating intracellular trafficking and processing of extracellular heparanase by mast cells (see reference 28 in the manuscript). The topic addressed is both interesting and significant, and the experiments are well-designed with adequate methods. The paper is informative, well-written, and supported by data that effectively contribute to the conclusions drawn. Additionally, the work is well-referenced.

However, there are a few points that could enhance the manuscript:

  1. In Figure 1, the investigation of L-Hpse and M-Hpse uptake into mast cells using ELISA is pivotal. To bolster the reliability of the findings, it would be beneficial to confirm the results with alternative methods such as immunofluorescent microscopy utilizing an anti-heparanase antibody. Additionally, visualizing the localization of heparanase in endosomes through colocalization experiments using immunofluorescent microscopy could provide valuable insights.

Response:
We agree with Reviewer 1’s comment. Particularly, immunofluorescent microscopy could provide information on both the internalized amount of Hpse and its intracellular localization. We have previously observed the fluorescent image of MST cells in which internalized L-Hpse was detected with granular distribution (ref.30).

In the revised manuscript, we have incorporated additional immunofluorescence data from internalized L- and H-Hpse in MST cells. This newly added Figure 1E shows that MST cells incubated with L-Hpse had a higher fluorescence intensity than MST cells incubated with M-Hpse. Also, the L-Hpse-dependent fluorescent signal was relatively concentrated in the intracellular area surrounding the nucleus, whereas the M-Hpse-dependent fluorescent signal was diffusely distributed throughout the entire cell body.

As colocalization experiments could provide further information, we have made attempts to investigate the earlier phase of internalization using double staining. However, we found that it was difficult because of the limited sensitivity of our confocal microscopy system. Provided we can secure the use of alternative instruments which fulfill the sensitivity required for our observation, we will continue this type of experiment.

In the revised manuscript, we have added the following statements to support the addition of Figure 1E:

Lines 125-129

Immunocytochemical images showed that the fluorescence intensity was higher in MST cells incubated with L-Hpse than in MST cells incubated with M-Hpse. The L-Hpse-dependent granular fluorescent signal was strong particularly in the intracellular area surrounding the nucleus of MST cells (Figure 1E).

Lines 137-140

(E) MST cells were incubated with L-Hpse and M-Hpse (400 ng/ml) for 8 h. Cell images im-munostained with anti-Hpse mAb (RIO-1), secondary antibody, and Alexa 568-labeled streptav-idin (Hpse). The cells were counterstained with Hoechst33342 (Hoechst). The right panels are merged figures of Hpse, Hoechst, and transmission images.

Lines 343-348

4.4. Immunocytochemistry.

MST cells were incubated with 400 ng/ml of Hpse protein for 8 h at 37°C. After the incubation, the cell suspension was subjected to cytospin, fixed with ice-cold methanol for 30 sec, stained with anti-muHpse mAb RIO-1, secondary antibody (biotinylated goat anti-rat IgG(H+L), Invitrogen), and Alexa568-conjugated streptavidin. The fluorescent cell image was taken using confocal microscopy (FV3000, Olympus, Tokyo, Japan).  

Lines 423-424

, Ms Yukina Shimojima (Hoshi University) for taking fluorescent images with confocal microscopy,

  1. There appear to be errors in the figure legend for Figure 1 (Page 3, lines 95-97): (A-C) MST cells were incubated with L- 95 Hpse (gray bars) and M-Hpse (black bars) (400 ng/ml) for the indicated periods, (B, C) PMC and BMMC were incubated with Hpse (500 ng/ml) for 6 h!!!!.

Response:
Thank you for pointing this out.  
We mistakenly wrote (A-C) instead of (A). The incubation conditions of MST and that of PMC/BMMC were different and should be distinctively described.

In the revised manuscript, we have corrected the statements as below,

Lines 131-134

(A-C) MST cells were incubated with L-Hpse (gray bars) and M-Hpse (black bars) (400 ng/ml) for the indicated periods, (B, C) PMC (B) and BMMC (C) were incubated with L-Hpse (gray bars) and M-Hpse (black bars) Hpse (500 ng/ml) for 6 h. In A-C, Hpse concentration in cell lysates was determined by ELISA.

  1. In Figure 3D, it is unclear how the control experiments were conducted, and this should be addressed for clarity and completeness.

Response:

Thank you for your comment. In the revised manuscript, we have added more explanation on how the experiment was conducted, and we hope that this will help to fully understand Figure 3D:

Lines 172-174

Hpse at a concentration of 100 ng/ml (gray bars) or 400 ng/ml (closed bars) was either pretreated with 1 mM of heparastatin (SF4) for 30 min (+SF4) or untreated (cont). These samples were and added to the ELISA plate.

Lines 364-374

Binding of Hpse to immobilized heparin or GAG derived from MST cells was examined based on an ELISA-like method as described previously [49,54]. Briefly, streptavidin was immobilized onto the ELISA plate (Greiner 655061). After blocking with 1% BSA-PBS, biotinylated GAG (10 µg/ml) in dilution buffer (PBS containing 0.1% BSA and 0.01% Tween 20) was added into each well. After washing out unbound GAG, Hpse was added into the wells and kept for 2 h at room temperature. In some experiments, Hpse was preincubated with heparastatin (SF4) before addition to the ELISA plate. Dilution buffer with or without heparastatin (SF4) was used as negative control. Binding of Hpse was quantified by coloration of ABTS at 405 nm. OD405 readings were normalized by subtracting the negative control OD values. The OD405 value of the negative control wells was 0.050 ± 0.003 (average ± S.D.).

  1. In the discussion section, it would be beneficial to discuss the potential role of other cell surface proteoglycans in the uptake of heparanase, as this could provide further insights into the mechanisms involved. Overall, addressing these points would strengthen the manuscript and contribute to a more comprehensive understanding of the topic.

Response:
We agree with Reviewer 1's comment. In a previous study, we have detected RNA expression of Sdc1, Sdc3 and Sdc4 in MST cells. Our immunocytochemical experiments indicated that expression of Sdc3 and Sdc4 was dominant, while the distribution of Sdc3 was likely inside the cells. Knockdown of the Sdc4 gene significantly decreased the uptake of Hpse, however, the suppression was approximately 50% [ref 30].

In the revised manuscript, we have incorporated the above thoughts as below,

Lines 227-231

Sdc4 gene knockdown experiments in our previous study strongly suggested the involvement of Sdc4 as a main receptor for the uptake of L-Hpse. However, since approximately 50% of the uptake remained after knockdown, we concluded that other cell surface GAGs such as Sdc1 and Sdc3 weakly expressed in MST cells [30] are also possibly involved in the uptake.

In addition, we have made minor corrections in the revised manuscript as follows,

  1. The affiliation of Tatsuro Irimura (affiliation 6) has been corrected to:

Division of Glycobiologics, Juntendo University Graduate School of Medicine; 2-1-1, Hongo, Bunkyo-ku, Tokyo 113-8421, Japan; t-irimura@juntendo.ac.jp

  1. Throughout the revised manuscript, we have made the use of the terms L-Hpse and M-Hpse more consistent.

  1. Lines 337-338

The lysates were subjected to sandwich ELISA as described elsewhere [29,30], and quantified as the concentration of Hpse in the cell lysates (ng/ml).

  1. Lines 422-423 (in response to the editorial office’s comment)

Conflicts of Interest: Dr. Motowo Nakajima is an employee of SBI Pharmaceuticals. All other authors declare no conflict of interest.

Sincerely yours,

Nobuaki Higashi

Reviewer 2 Report

Comments and Suggestions for Authors

The current manuscript entitled "Uptake of heparanase into mast cells is regulated by its enzymatic activity to degrade heparan sulfate" addresses an interesting and relevant biological question about the regulation of heparanase uptake by mast cells and how its enzymatic activity influences this process. The present manuscript presents a detailed and well-structured experimental approach, including the use of various mast cell types (MST cells, PMC, and BMMC) and different forms of heparanase (L-Hpse and M-Hpse). The study employs a range of techniques such as ELISA and flow cytometry, which are appropriate for investigating the uptake and binding of heparanase. However, I suggest certain changes that should be addressed in the revised manuscript.

1. Abstract: Please provide quantitative parameters of the findings.

2. Introduction: It lacks depth. Provide a more comprehensive review of the literature on heparanase and mast cell biology.

3. Methods: Please include a section of statistical analyses.

4. DIscussion: While the discussion addresses the study's findings, it could be expanded to include the broader implications of the research, potential applications, and future directions.

5. Mention the significance of the results and their contribution to the field.

Author Response

May 29, 2024

Response to Reviewer 2 Comments

[IJMS] Manuscript ID:  ijms-3025655 : major revisions
manuscript ID:  ijms-3025655
Type of manuscript: article
Title: Uptake of heparanase into mast cells is regulated by its enzymatic activity to degrade heparan sulfate.
Section: Macromolecules
Submitted to special issue: Heparin, Heparan Sulfate and Heparanase in Health and Disease

Dear Editors,

First of all, let me express my sincere thanks for taking the time to review our manuscript and for your encouragement to submit a revised version. We carefully read the Reviewers` comments and hereby submit our point-by-point responses, including an explanation on how the manuscript was revised. In the revised manuscript, major revisions are indicated in red font. We hope that the revisions are satisfactory for the Reviewers.

Reviewer 2:
The current manuscript entitled "Uptake of heparanase into mast cells is regulated by its enzymatic activity to degrade heparan sulfate" addresses an interesting and relevant biological question about the regulation of heparanase uptake by mast cells and how its enzymatic activity influences this process. The present manuscript presents a detailed and well-structured experimental approach, including the use of various mast cell types (MST cells, PMC, and BMMC) and different forms of heparanase (L-Hpse and M-Hpse). The study employs a range of techniques such as ELISA and flow cytometry, which are appropriate for investigating the uptake and binding of heparanase. However, I suggest certain changes that should be addressed in the revised manuscript.

  1. Abstract: Please provide quantitative parameters of the findings.

Response:
Thank you for your comment. 
As Reviewer 2 commented, the original Abstract included qualitative descriptions, such as “The uptake of M-Hpse into mast cells was increased”, “The uptake efficiency of mutated M-Hpse was significantly higher” etc. In the revised Abstract, we have replaced these phrases with quantitative ones and made additional revisions as follows. Words with strikethrough are deleted from the text of the revised manuscript.

Lines 27-28

The internalized amount of L-Hpse was significantly higher than that of M-Hpse.  

Lines 31

The internalized amount uptake of M-Hpse into mast cells was significantly increased...  

Lines 34

The internalized amount uptake efficiency of mutated M-Hpse was...  

  1. Introduction: It lacks depth. Provide a more comprehensive review of the literature on heparanase and mast cell biology.

Response:
Thank you for your suggestion. 

In the revised manuscript, we have substantially expanded the Introduction and re-arranged the contents as follows, with the underlined parts being newly added.

(1) Hpse: functional differences of latent- and mature-form Hpse, (2) significance of Hpse in many inflammatory diseases, (3) mast cells and their secretory granules, (4) uptake of extracellular molecules in the mast cells (a large part of the description was moved from the Discussion part), (5) possible involvement of Hpse in mast cell migration, (6) possible involvement of Hpse in processing of macromolecular heparin of mast cells, and (7) uptake of Hpse in mast cells shown in our previous study and the aim of the study, i.e., to clarify whether the uptake process merely replenishes the granular enzyme, or it is associated with the maturation of Hpse.

To make the manuscript comprehensive, it is proper to cite the important literatures available in the research field. In Pubmed, the keyword search (heparanase and mast[title]) resulted in seven hits. Four out of these seven were already cited in the original manuscript [ref 24,26,29,30]. In the revised manuscript, we have added one more reference [48]. With this addition we believe that the revised manuscript properly covers the information on mast cells and heparanase currently available.

In the revised manuscript, we have elaborated on how heparanase is involved in the progression of inflammatory diseases. This should strengthen the significance of mast cells as a potential producer of inflammatory Hpse. We have incorporated seven references [15-21] on Hpse expression in inflammatory diseases, one reference [25] on Hpse expression in circulating cells, three references [42-44] on migration of mast cells in inflammatory diseases, and one reference [48] on the effect of Hpse on mast cell function.

Lines 55-60

Hpse upregulation has been reported in many inflammatory diseases including delayed-type hypersensitivity, psoriasis, inflammatory bowel disease, rheumatoid arthritis, atherosclerosis, fibrosis, pulmonary inflammation, pancreatitis, acute kidney injury, etc [15-22]. Comprehensive gene expression analysis shows relatively high expression of Hpse in leukocytes including mast cells, which has been confirmed by us and other researchers [23-26]. In addition, epithelial cells can produce Hpse in inflammatory diseases.

Lines 61-78

Mast cells trigger allergic and inflammatory reactions by releasing various vasoactive and inflammatory substances. Secretory granules of mast cells contain a wide spectrum of substances, such as vasoactive amines, cytokines, lysosomal enzymes, etc. A large part of them are preformed and stored in the granules. In addition, mast cells can take up extracellular materials and transfer them into the secretory granules. The uptake process is required for the maturation and activation of the internalized proteins. As an example, internalized TNF-alpha becomes mature during the uptake process [27,28].

Hpse is essential for heparin processing in mast cells. It is likely that mast cells produce latent-form Hpse, secrete it, and then take it up and convert it into mature-form Hpse for use. Since mast cells release the mature-form Hpse as a result of degranulation, replenishment of the enzymatically active Hpse in the granules is essential for maintaining continuous heparin processing. Our previous study indicated that recombinant latent-form Hpse can be internalized and sorted into secretory granules. The enzyme was processed into mature-form Hpse during the internalization process [29,30]. Additionally, this process is relevant to replenish granular contents that has been released via degranulation. In this case, endocytosis occurring coordinately with the degranulation has a role in the replenishment [31]. It has been reported that mast cells can internalize IL-17, MHC class II, PSGL-1, and IgE fragments presumably associated with FceRI [32-36]. Despite these studies, the molecular mechanism of the endocytic process in mast cells has not been intensively studied.

Lines 88-90

Although mast cells also migrate into inflammatory lesions in diseases such as allergic airway inflammation and atherosclerosis [42-44],

Lines 98-102

Other researchers indicated that Hpse expression results in decreased storage amounts of granular enzymes such as carboxypeptidase A and mMCP-5 [26], as well as in increased production of total GAGs [48].

The above two aspects are related to the expression level and maturation status of Hpse in secretory granules.

Lines 104-112

Our precedent study proposed a possible pathway, that is, mast cells take up extracellular latent-form Hpse and store it in their secretory granules. During this process, the internalized Hpse was processed into mature-form Hpse that can degrade macromolecular heparin in the secretory granules into fragments. Syndecan 4 (Sdc4) on the mast cell surface is considered essential and the dominant surface molecule for Hpse uptake [30]. The aforementioned study, however, did not mention whether the uptake process merely replenishes the granular enzyme, or whether it is associated with the maturation of Hpse.

  1. Methods: Please include a section of statistical analyses.

Response:
Thank you for your valuable comment. 
In the original manuscript, a statistical analysis statement was included in "4.3. Quantification of Hpse uptake in mast cells."  In the revised manuscript, we have moved this statement to the section “4.9. Statistical analysis.”.

Lines 398-400

4.9. Statistical analysis.

The significance of differences in the data was evaluated using the two-tailed Student’s t-test.

  1. DIscussion: While the discussion addresses the study's findings, it could be expanded to include the broader implications of the research, potential applications, and future directions.

Response:
Thank you for your suggestion. 
To discuss the broader implications of the research, we elaborated on three points in the revised manuscript; (1) scavenge of latent-form Hpse to avoid excessive inflammation, (2) possible involvement in uptake of other heparan sulfate-associated cytokines such as FGF and VEGF, and (3) mature-form Hpse-mediated unresponsiveness to retard the uptake process.

In (2), we have incorporated four references [58-61] on the expression of FGF and VEGF in the secretory granules of mast cells. In (3), it is possible that this unresponsiveness may sustain the inflammatory condition because the extracellular inflammatory molecules are not rapidly removed in the extracellular space of degranulated mast cells. For potential application, administration of a Hpse inhibitor like heparastatin (SF4) possibly accelerates the uptake process of the mast cells, which could be possibly therapeutically relevant. Regarding the future directions, we have amended some statements in the last part of the Discussion. In the following, we have listed all the revisions which we have made.

Lines 197-200

An essential question in the present study was whether the uptake process of Hpse merely replenishes the granular enzyme, or whether it is always associated with the maturation of Hpse. If the latter is the case, the underlying mechanism how L-Hpse is preferentially internalized becomes an additional question. 

Lines 209-210

Taken together, we propose a regulatory role of Hpse enzymatic activity for the preferential uptake of L-Hpse.

Lines 267-284

The preferential uptake of latent-form Hpse likely facilitates efficient maturation of Hpse. Although the further physiological relevance of the uptake of Hpse remains to be elucidated, implications of the preferential uptake could be discussed focusing on three points. First, the uptake system is beneficial to avoid exaggerated inflammation induced by latent-form Hpse [10,14,57]. In other words, scavenging of latent-form Hpse may be an anti-inflammatory action by mast cells. Second, because there are many kinds of biologically active molecules that can bind to heparan sulfate, the mature-form Hpse-mediated regulation is also possibly suppressive for the uptake of other HS-associated molecules. We are currently working on clarifying this point by focusing on the heparin-binding cytokines FGF and VEGF that are stored in the secretory granules of mast cells [58-61]. Third, the presence of mature-form Hpse in the extracellular space may confer unresponsiveness to mast cells. We and others reported that secretory granules accumulate mature-form Hpse [23-26]. When this Hpse is released, degradation of cell surface HS by the released Hpse possibly retards the uptake process of mast cells. If such degradation did not occur, mast cells could rapidly re-internalize the released materials, which may decline mast cell-derived inflammatory responses. It is interesting that this process can be pharmacologically accelerated by addition of Hpse inhibitors like heparastatin (SF4) (Figure 3). Further studies on the manipulation of mast cell-mediated uptake are ongoing. 

  1. Mention the significance of the results and their contribution to the field.

Response:
Thank you for your comment. 
Regarding the significance of the research, this is the first research to report a possible regulatory mechanism of Hpse to accelerate the maturation of Hpse, also to scavenge the latent-form Hpse from inflammatory environments. This regulatory mechanism possibly extends to the uptake of other HS-associated molecules such as FGF and VEGF. Our findings contribute to the inflammation research field, by providing a novel strategy to regulate extracellular inflammatory conditions.   

In the revised manuscript, we have amended the Conclusions as follows,

Lines 402-408

In the present study, we propose a novel role for the enzymatic activity of Hpse to regulate the uptake efficiency of Hpse into mast cell granules. This regulatory system is potentially significant for accelerating the maturation of Hpse enzyme and for scavenging latent-form Hpse from inflammatory environments. Our findings imply that mature-form Hpse-mediated suppression of the uptake can be manipulated by using heparanase inhibitors, which can potentially provide a novel strategy to regulate extracellular inflammatory conditions.    

In addition, we would like to make minor corrections on the manuscript as follows, if permitted.

  1. The affiliation of Tatsuro Irimura (affiliation 6) has been corrected to:

Division of Glycobiologics, Juntendo University Graduate School of Medicine; 2-1-1, Hongo, Bunkyo-ku, Tokyo 113-8421, Japan; t-irimura@juntendo.ac.jp

  1. Throughout the revised manuscript, we have made the use of the terms L-Hpse and M-Hpse more consistent.

  1. Lines 338-339

The lysates were subjected to sandwich ELISA as described elsewhere [29,30], and quantified as the concentration of Hpse in the cell lysates (ng/ml).

  1. Lines 423-424 (Response to the editorial office’s comment)

Conflicts of Interest: Dr. Motowo Nakajima is an employee of SBI Pharmaceuticals. All other authors declare no conflict of interest.

Sincerely yours,

Nobuaki Higashi

Round 2

Reviewer 2 Report

Comments and Suggestions for Authors

The authors have satisfactorily addressed all comments in the revised manuscript.